# Ethanol Kinetics in the Human Brain Determined by Magnetic Resonance Spectroscopy

**DOI:** 10.3390/ijms241713499

**Published:** 2023-08-31

**Authors:** Annette Thierauf-Emberger, Dominik Schuldis, Michael Dacko, Thomas Lange

**Affiliations:** 1Institute of Forensic Medicine, Medical Center—University of Freiburg, Faculty of Medicine, University of Freiburg, 79104 Freiburg, Germany; dominik.schuldis@uniklinik-freiburg.de; 2Division of Medical Physics, Department of Diagnostic and Interventional Radiology, Medical Center—University of Freiburg, Faculty of Medicine, University of Freiburg, 79106 Freiburg, Germany; dacko.michael@gmail.com (M.D.); thomas.lange@uniklinik-freiburg.de (T.L.)

**Keywords:** brain, ethanol, kinetics, magnetic resonance spectroscopy

## Abstract

In many parts of the world, ethanol is a widely consumed substance that displays its effect in the brain, the target organ for desired, but also negative impact. In a previous study, the ethanol concentrations were analyzed in different regions of the brain by magnetic resonance spectroscopy (MRS). In this study, the same method is used to demonstrate the kinetics of the ethanol concentration in the human brain after oral ethanol uptake. A drinking study was performed with 10 healthy participants. After the uptake of ethanol in a calculated amount leading to a plasma ethanol concentration of 0.92 g/L (19.95 mM corresponding to a blood ethanol concentration of 0.7 g/kg), brain ethanol concentrations were continuously measured by means of MRS on a 3 Tesla human magnetic resonance imaging (MRI) system. For the data acquisition a single-voxel sLASER sequence was used, with the volume of interest located in the occipital cortex. Intermittently, blood samples were taken and plasma was analyzed for ethanol using headspace gas chromatography with flame ionization detection (HS-GC-FID). The obtained MRS brain ethanol curves showed distinct inter-individual differences; however, a good intra-individual correlation of plasma and brain ethanol concentrations was observed. The results suggest a rapid equilibration between blood and brain. The ethanol concentrations measured in the brain were substantially lower than the measured plasma ethanol results, suggesting an MRS visibility of about 63% for ethanol in brain tissue. The maximum individual ethanol concentrations in the brain (normalized to water content) ranged between 7.1 and 14.1 mM across the cohort, while the highest measured plasma concentrations were in the range between 0.35 g/L (9.41 mM) and 0.95 g/L (20.52 mM).

## 1. Introduction

Even though severe negative effects are sufficiently known, ethanol is still a largely consumed substance in many parts of the world. The negative and positive impact of ethanol is associated with the brain, the primary target organ for ethanol effects. The positive consequences of ethanol consumption are, e.g., euphoria, relaxation, and an increased feeling of self-worth [1]. Particularly in higher concentrations, this desired effect is opposed by disinhibition, aggression, cognitive and motoric impairments up to violence, road traffic accidents, coma, and death on the one hand, and alcohol abuse disorders and dependence on the other hand [1].

The onset is rapid, which is explained by the small size of the ethanol molecule, the rapid diffusion and distribution within the aqueous structures of the body, and the permeation of the blood–brain barrier [2].

The molecular causation of the ethanol effects has long been the focus of the research. Ethanol distributes within the body and brain similar to water, and reaches an equilibration within the brain and its cells within a few minutes of drinking [2]. In the brain, ethanol alters the function across a wide range of concentrations and phases of drinking [2]. The ethanol molecule interacts with other biomolecules via hydrogen bonding and weak hydrophobic interactions and has a limited potential [2]. It acts on many cellular and molecular targets and influences the regulation of neuronal communication within the brain [3,4,5]. Ethanol has an effect on synaptic targets, such as ion channels, neurotransmitter receptors, and intracellular signaling proteins [3]. The activity of neuronal circuits is affected by ethanol on different levels [6].

For clinical and forensic purposes, ethanol concentrations are mainly determined in the serum and blood and do not provide information about brain ethanol values. Brain ethanol concentrations can be measured in postmortem samples or by proton magnetic resonance spectroscopy (MRS). Several postmortem studies address the correlation between blood and brain ethanol concentrations; however, they show differing results [7,8,9,10,11]. These postmortem data obviously lack kinetic observations.

Studies on ethanol by means of MRS also focus on the comparison of blood and brain ethanol concentrations [12,13,14], but also on measurement techniques [12,15,16] and detectable changes due to frequent or chronic ethanol consumption [15,17,18]. MRS research on the comparison of blood and brain ethanol concentrations has, to date, not resulted in consistent ratios. Few studies were performed on living humans, primates, and rats and showed higher, similar, or lower blood than brain concentrations [12,13,17]. The results achieved by MRS mainly depend on the applied method, above all on the reference metabolite, and the tissue composition [13,17,19], but also on the estimated T2 relaxation constant of ethanol and the reference metabolite. For the differences between the matrices, an invisible ethanol pool with extremely short T2 relaxation, arising from the interaction of ethanol with membrane lipids, is held responsible [17,20]. Only very few data with regard to kinetic parameters have been published [12].

In a previous study, this working group performed a drinking experiment to compare the ethanol concentrations in four different regions of the brain as well as the serum and brain concentrations [21]. The brain concentrations were successively measured by MRS in the frontal and occipital cortices, the putamen, and the cerebellum, while the serum concentrations were observed by a forensic routine method (headspace gas chromatography-flame ionization detection, HS-GS-FID). The measurement accuracy of the chemical method by far exceeded the MRS measurement accuracy. The ethanol concentrations in the four anatomical regions were quite similar; slight differences could be explained by the time delay and limited accuracy. The proportion of cerebrospinal fluid and vascularization was considered by normalization to the water content of the voxel. The determined serum ethanol concentrations were much higher than the corresponding brain MRS values at nearly all measuring points.

In view of the knowledge that the concentrations within the brain are comparable, the considerations focus on the kinetics of the ethanol values in the brain. It is the aim of this drinking study to determine the ethanol time course in the brain in a realistic scenario with oral uptake and to depict the correlation of the strong ethanol effects shortly after drinking. Furthermore, we aim to compare the curve characteristics between the brain and serum ethanol concentrations. To date, only Hetherington et al. as well as Kubo et al. have presented some kinetic data on this topic and performed the comparison between the blood and the brain of the subject under discussion [12,14]. For Hetherington et al., however, it has to be noted that the matrix for the MRS measurement was not the brain tissue within the voxel, but the aqueous components of the measurement voxel [12]. In both studies [12,14], the observation period was far shorter and more interrupted. Due to the rapid diffusion, distribution, and equilibration, it can be hypothesized that the brain ethanol kinetics closely follow the serum ethanol curve. This hypothesis is addressed by the present study and has not been evidenced over a wide range of the ethanol curve before.

## 2. Results

The subjects declared to have remained abstinent, and the blood and breath ethanol concentrations were all negative prior to drinking. The period of drinking took between 6 (V10) and 37 (V5) min (average 18.5 min). The plasma ethanol concentrations of the samples obtained at the end of the drinking period showed great differences and ranged between 0.04 g/L (0.86 mM, V10) and 0.95 g/L (20.52 mM, V7) (average 0.38 g/L, 8.6 mM). The subsequent blood sampling occurred during test realisation time about every 50 min for approximately 2.5 h, even though sampling intervals of 30 min were planned.

Figure 1 presents the MR spectrum (as processed by LCModel) from the additional measurement acquired at the beginning of the first MRS session, shortly after the end of the drinking period. In addition to the characteristic Eth resonance, prominent peaks from the total N-acetyl aspartate (NAA), total creatine (Cr), and choline-containing compounds (Cho) can be distinguished. The quantification results for several major brain metabolites and ethanol (Eth) can be seen on the right side along with standard error estimates in the form of Cramér-Rao lower bounds (CRLBs) [22].

The blood and brain ethanol concentrations (normalized to the water fractions of the blood and brain, respectively) of the 10 participants are presented in Figure 2. Note that, for subject V9, the block MRS2 had to be aborted due to a biobreak. V09 had to be taken out of the scanner and MRS2 was performed as the first block of the second session. The highest measured blood and brain ethanol concentrations are listed in Table 1 for the individual subjects. The brain ethanol concentrations show considerable differences in the ten participants; however, also the curve progression varies substantially. In the first MRS session, the maximum concentrations ranged between 7.1 mM (V3) and 14.0 mM (V2). When taking the linearly fitted MRS ethanol concentration curves into account, the ascending part of the curve was missed for subjects V1, V2, V3, V4, V5, and V7. The curve of V10 presents a steep rise during MRS1 with a total increase of approximately 2.9 mM. In contrast, the linear fit of MRS2 exhibited a substantial decrease, suggesting that the ethanol peak was missed during the measurement break. At the end of the MRS acquisition (140 to 180 min after the start of the MRS measurement), the brain ethanol concentrations ranged between 1.1 and 5.7 mM.

The blood ethanol concentration values and brain ethanol concentration curve (both normalized to the water fraction of the blood and brain, respectively) of the additional measurement are presented in Figure 3.

The relationship between the plasma and brain ethanol concentrations across all ten subjects are presented in Figure 4a. The correlation analysis yields a Pearson’s correlation factor of 0.93 across the cohort. Figure 4b presents the linear relationship between the plasma and brain ethanol concentrations of the additional measurement (slope = 0.49, intercept = −0.62). The correlation coefficient of 0.9994 suggests an almost perfect linearity between the plasma and brain concentrations of ethanol. Figure 4c presents the correlation of V10, including an outlier. The details of the intra-individual regression analysis (slope, intercept, and correlation coefficient) for the ten subjects can be observed in Table 2. It is striking that V10 only shows a correlation coefficient of 0.48, while it varies between 0.97 and 1.0 for all other subjects. The linear regression graph (Figure 4c) of V10 suggests that there is a strong discrepancy between the plasma ethanol concentration and corresponding brain concentration for the second blood sample obtained immediately after MRS1.

## 3. Discussion

This study addressed the kinetics of ethanol in the brain in comparison to the plasma ethanol concentration and represents a new approach to the connection of MRS and the detectability of alcohol. According to the hypothesis, there is a good agreement between the brain ethanol curve and plasma ethanol analytical values. The point values of serum ethanol concentration fit in the (interrupted) MRS ethanol curves from the brain very well. The informative value of the single plasma ethanol concentrations was limited, however, due to the time intervals between the sampling. This mainly affected the ascending and peak part of the curve; the peak or plateau could be missed by the sampling procedure. Special emphasis was placed on the MRS analyses, which had to be interrupted for blood sampling. Even though it was planned to take blood samples approximately every 30 min, the first measurement interruption and blood sampling only occurred 47 to 61 min after the end of drinking. In most of our test persons, the peak concentration in plasma was assumedly achieved within this time period.

In V6, V8, and V10, the combined evaluation of the plasma ethanol concentrations and MRS brain ethanol curves demonstrated a late onset of the peak or plateau of the curves. In these three test persons, the relevant parts of the ascending curve could be captured by the MRS measurements. A closer examination of these three participants did not provide a clear explanation of the similar curve progression. As demonstrated in Table 3, V6 is a 67-year-old male participant with arterial hypertension and a regular intake of Ramipril. V8 is a 29-year-old female person without any known illnesses and medication. V10 is a 61-year-old male suffering from tinnitus and regularly applying testosterone. The body mass index was normal in all three individuals. There was an intraindividual correlation between the maximum brain and serum concentrations; however, there was no interindividual comparability: V10 reached rather high concentrations in both matrices, while the results of V8 were at the lower end of the observed range. V10 showed the shortest drinking time by far and the shortest interval between the beginning of drinking and the first MRS measurement. This can be a plausible explanation for the most complete presentation of the MRS curve. This was not applicable to V6 and V8 with average durations of the drinking period and the interval to the first measurement (V6: 29 and 56 min, respectively; V8: 15 and 34 min, respectively). There was no deducible influence of age, and an effect of the aforementioned medication was unlikely. Among all the subjects, only for V7 could the influence of the medication (Ramipril, hydrochlorothiazide, and amlodipine) be discussed since gastric side effects were described for the combination product of Ramipril and amlodipine (disturbed gastric and intestinal draining); however, the MRS curve of this subject did not show this abnormality. The cohort results do not provide a clear indication of the reason for the interindividually different curve progression; however, they depict the close correlation of plasma and brain ethanol concentrations. An explanation for the differing curves has to be searched in individual factors, such as gastric and intestinal motility and blood circulation, because all the participants consumed the same beverage some hours after eating a light meal [23]. Apart from V7, no medical reasons were provided for a possibly abnormal ethanol diffusion.

In accordance with other MRS and several postmortem studies, the comparable concentrations in the serum were higher than in the brain [7,8,9,10,12,21]. Bonventre et al. [8] observed such a close correlation that a formula to calculate the brain concentration was suggested as follows: C_brain_ (g/100 g) = 0.487 C_blood_ (g/dL) + 0.055. The concentration ratio between the blood and brain observed by Bonventre et al. was 0.97 for blood levels of 0.1 g/dL and 1.32 for blood levels of 0.2 g/dL [8]. However, the formula has not achieved broad consensus. Chiu et al. [20] observed significantly higher brain–blood ethanol concentration ratios for heavy drinkers than for occasional drinkers. In this study, the serum concentrations substantially exceeded the brain concentrations in all subjects and at all times. It should be noted that, in MRS studies, the ethanol results strongly depend on the measurement parameters and quantification methods (e.g., the reference metabolite). In particular, the T2 relaxation constants used for correction had a substantial influence on the quantification results, and the literature values ranged between T2 = 335 ms measured at 1.5T [24] and T2 = 82 ms measured at 4T [16].

An overall comparison of the plasma (C_plasma_) and brain ethanol concentrations (C_brain_) measured in this study yielded a very strong correlation (r = 0.94), which was higher than the correlation coefficients reported by Fein and Meyerhoff (r = 0.58); although, their study used a substantially longer acquisition time of 4 min per spectrum compared to the 96 s used in our work [25]. The correlation coefficients reported by Kubo et al. for various acquisition times (30/58/106/298 s) were also slightly lower than in our study [14]. Interestingly, their study showed an increase in the correlation coefficient with acquisition times between 30 and 106 s, but no further enhancement when the acquisition time was increased to 5 min. These results suggest that an acquisition time of ~100 s as chosen in our study represents a reasonable compromise between the signal-to-noise ratio and temporal resolution. Moreover, it was remarkable that such a strong correlation was observed, despite the heterogeneity of our subject cohort (both sexes; large age range). This represents the most significant finding of this study. As previously mentioned, the brain is the target organ for the effect of ethanol; however, for the quantification, regularly peripheral venous blood is used. The correlating kinetics justify the validity of this method. A routine use of spectroscopic techniques for diagnostic purposes of acute ethanol impairment is not worth considering for several reasons, in particular, the high expense of this method on the equipment as well as on the time exposure.

The following linear relationship between the two measures was obtained: C_brain_ = 0.63·C_plasma_ − 0.26. This suggests a MRS ethanol visibility of about 63%, which is in the midrange of values reported in the literature [12,13,17,20,26,27,28,29]. The negative intercept might have occurred due to the fact that the MRS ethanol resonance was partly modeled by the baseline of the LCModel fit, giving rise to a slight underestimation of brain ethanol concentrations.

In contrast to the other subjects and especially the extra measurement, V10 showed a much weaker correlation between the measured blood and brain ethanol concentrations due to an outlier observed for the third blood sample. The serum ethanol measurements were repeated with comparable results, and the labeling of the samples was performed according to the dual control principle. A plausible explanation for the serum ethanol concentration outlier in V10 was not apparent.

The presented study had some limitations. On the technical side, the greatest bias may have been introduced by the imperfect relaxation correction. Ideally, a very short TE and a very long TR should be chosen for the MRS acquisition to minimize this bias. We chose a TE = 74 ms since it yielded smaller error estimates (CRLBs) for the ethanol quantification compared to the minimal echo time of TE = 40 ms, which would be feasible with the used sLASER implementation. The chosen TR = 1.5 s is a commonly used setting in single-voxel MRS studies at 3T and represents a compromise between the optimal SNR and minimization of the T1 bias. While a T2 relaxation correction could be based on the literature values, albeit obtained at a slightly different field strength, no T1 relaxation values were available from previous MRS studies. However, we expect the T2 relaxation to give rise to a much larger quantification bias than the T1 relaxation for the given TE/TR setting. It should be noted that a T1 bias should be smaller for ethanol than for other major brain metabolites, which are only present in grey matter (GM) and white matter (WM), but not in cerebrospinal fluid (CSF). Ethanol is also present in CSF and, due to its excellent water solubility, the ethanol concentration scales with the tissue water fraction. In terms of the T1 bias, the tissue composition (GM/WM/CSF) will therefore affect the quantification results for the ethanol and water reference signals in a similar way, since it can be assumed that ethanol T1 in CSF is much longer than ethanol T1 in GM and WM (as is the case for water T1). In addition to the relaxation bias, it was mainly the unresolved issue of ethanol visibility in the brain that contributed to the quantification uncertainty. It has been hypothesized that ethanol is not only present in intra- and extracellular fluids where it is MR-visible with standard MRS sequences, but also exists in a cell membrane pool where its molecular mobility is strongly restricted by the interaction with macromolecules, giving rise to extremely rapid T2 relaxation. This hypothesis was supported by studies using an MRS sequence with magnetization transfer saturation pulses [28]. This immobile ethanol pool may be inaccessible by MRS techniques, which are based on localization methods, such as point resolved spectroscopy (PRESS), stimulated echo acquisition mode (STEAM), semi-adiabatic localization by adiabatic selective refocusing (sLASER), and variants thereof. A dedicated ultra-short-TE spectroscopic imaging method might be able to make this pool MR-visible; however, the resulting ethanol resonances would be so broad that they could not be distinguished from the background of other signals with extremely short T2. It should be noted that accidental magnetization transfer effects, e.g., arising from the water suppression module, might also reduce the ethanol visibility of the MR-visible ethanol pool. To minimize this effect, frequency-selective water suppression pulses with a small bandwidth of 50 Hz were applied in this study; however, an influence on the quantification results cannot be entirely ruled out.

In terms of the overall study design, the sampling regime did not stick to the study plan due to technical circumstances. The MRS measurement had some lead time, which has to be considered in future study designs. To provide an outlook, an additional subject was surveyed with the same method; however, fewer and shorter interruptions and a substantially longer MRS acquisition (with three additional MRS blocks) to obtain a more complete brain ethanol curve were employed. Secondly, only two participants (V2: 0.92 g/L (19.95 mM); V7: 0.95 g/L, 20.52 mM) achieved the aimed plasma ethanol concentration of 0.92 g/L (19.95 mM). Some of the other subjects presented maximum concentrations of about half of the target concentration (V3: 0.43 g/L (9.40 mM); V8: 0.50 g/L (10.83 mM)). This might have given rise to an MRS quantification bias associated with the signal-to-noise ratio. In additional projects, the calculation of the drinking amount (by the Widmark formula) has to be adopted. It also has to be noted that the generalization of the results of this study is restricted by the small number of participants.

## 4. Materials and Methods

### 4.1. Experimental Set-Up

The study was approved by the Ethics Committee of Freiburg University (project nr. 342/20). All study participants provided their written informed consent after the complete description of the study was delivered to the test subjects. The subjects were recruited via notice boards.

Ten healthy subjects (V = volunteer) participated in this study (5 male, 5 female). The participants intentionally presented a wide age range (28 to 67 years old). Personal data, including the state of health, are shown in Table 3, while the data regarding the experimental parameters are presented in Table 1. All participants stated having balanced nutrition; V4 had a gluten-free diet due to coeliac disease.

Drinking experiments were performed as follows: after at least 2 days of abstinence from alcoholic beverages and at least 2 h after a light meal (bread roll with low-fat topping), the ethanol zero value was verified by a void blood sample and the measurement of the breath ethanol concentration. The breath ethanol concentration was determined using a mobile handset (Draeger Alcotest 6510, Lübeck, Germany). Each volunteer then drank an individually calculated amount of wodka (40 vol-%, optionally diluted with lemonade) within 30 min, aiming at a plasma ethanol concentration of 0.92 g/L (19.95 mM, corresponding to a blood ethanol concentration of 0.7 g/kg). The calculation was based on Widmark’s equation. At the end of the drinking period, a blood sample was obtained. For better comparability with the brain concentrations, the ethanol concentration was related to the water fraction of blood.

Then, the subjects were positioned in the MR scanner. After the acquisition of an anatomical dataset for voxel positioning (duration of approximately 7 min), the MRS measurements were continuously performed in the occipital cortex for about 2.5 h. The occipital cortex was chosen as a large homogeneous brain region where robust ethanol detection could be demonstrated in a previous study [21]. The measurements were interrupted for a short functional MRI (fMRI) sequence (6 min, to be described elsewhere), blood sampling, and visits to the toilet. It was planned to take the blood samples at the following points of time: 0.5, 1, 1.5, 2, and 2.5 h after the end of the drinking period. After 2.5 h of measuring, the breath ethanol concentration was repeatedly determined. Once the test persons showed less than 0.15 mg/L of breath ethanol, they were allowed to leave the experimental setting.

### 4.2. Chemicals and Instrumentation

#### 4.2.1. Determination of the Blood Ethanol Concentration

For blood sampling, safety cannulas (21 G) and Monovette (S-Monovette^®^ 9 mL, serum with a clot activator from Sarstedt (Nümbrecht, Germany)) were used. Serum ethanol concentrations were measured using headspace gas chromatography-flame ionization detection (HS-GC-FID) with t-butanol as the internal standard. Ethanol determination was performed using a linear calibration with aqueous calibrators containing 0.1, 0.2, 0.5, 1, 2, 3, 4, and 5 g/L of ethanol. The lower limit of quantitation (LLOQ) was the lowest calibrator’s concentration (0.11 g/L (2.28 mM) for plasma or 0.08 g/kg for blood). The method used was fully validated. For a better comparability of the results, the plasma ethanol concentration was preferred over the blood ethanol concentration in this study.

#### 4.2.2. MRS Measurements

The MRI and MRS measurements were performed with a 3T Prisma MR system (Siemens Healthineers, Erlangen, Germany), using a 64-channel receive coil. The whole MRS protocol consisted of two scan sessions interrupted by a short bio break (Figure 5a). Blood samples were obtained right before the first session, right after the second session, during the session break, and in the middle of each session. For the two blood samples obtained during the sessions, the subject stayed on the scanner bed to avoid the repositioning of the MRS volume of interest (VOI). Thus, two MRS measurement blocks were performed per session (1st session: MRS1 + MRS2; 2nd session: MRS3 + MRS4). At the beginning of each session, an anatomical measurement with a T1-weighted magnetization-prepared rapid gradient echo (MP-RAGE) protocol was conducted for the positioning of the MRS VOI and tissue segmentation for absolute quantification. Magnetic resonance spectroscopy data were acquired with single voxel semi-adiabatic localization by adiabatic selective refocusing (sLASER) [30] from a quadratic VOI (19.7 mL) located in the occipital cortex (Figure 6). The sLASER protocol contained hyperbolic secant pulses with a bandwidth of 5 kHz for slice-selective refocusing and used a repetition time TR = 1.5 s and an echo time TE = 74 ms. This echo time was chosen to yield a symmetric triplet resonance with a maximum in-phase signal for ethanol, with a strong positive central peak and two weaker inverted outer peaks. Water suppression was performed with a water suppression enhanced through T1 effects (WET) module consisting of three water-selective saturation pulses with a bandwidth of 50 Hz and subsequent spoiler gradients [31]. Additionally, at the end of the first session, a short water-unsuppressed reference scan was performed for absolute quantification via the internal water reference method [32]. For a more thorough assessment of the ethanol degradation curve, an additional longer MRS measurement was performed on another healthy subject (male, 33 years old, 86 kg, 191 cm, drinking time 24 min). During this extra measurement, no fMRI data were acquired and the MRS protocol contained 7 blocks of 30 min each (Figure 5b), interleaved with 4 blood samples obtained during the measurement and at the end of the session.

#### 4.2.3. Postprocessing and Quantification

Spectral fitting and metabolite quantification were performed with the analysis software LCModel (version 6.3-1R) [33], using numerically simulated basis spectra for ethanol and 18 brain metabolites, which are typically detectable in vivo. Numerical simulations were performed with the PyGAMMA library from the VeSPA package, using shaped RF waveforms for the refocusing pulses [34]. From the anatomical dataset, GM, WM, and CSF tissue fractions of the measurement voxel were determined via segmentation with Freesurfer software (version 7.1.1) [35]. These tissue fractions were used for estimating the water content of the MRS measurement voxels, assuming water concentrations of 42.9 mol/L (water fraction: 78%) for GM, 35.8 mol/L (water fraction: 65%) for WM, and 53.4 mol/L (water fraction: 97%) for CSF. For fitting and quantification, the continuously acquired MRS data were divided into batches of 64 spectral averages, which amounted to a temporal resolution of 96 s. For the sake of robustness, absolute quantification (of ethanol and Cr) was only performed for the first spectrum/time point of each MRS measurement. For all other spectra, the ethanol concentration was referenced to Cr, which was assumed to remain constant throughout the experiment. This procedure turned out to be more robust than the direct absolute quantification of every individual spectrum. It was verified by assessing NAA concentration fluctuations (unrelated to the ethanol level) in the time courses, which could be avoided with this method.

The measured ethanol concentrations were corrected for T2 relaxation effects, using the ethanol T2 constant determined at a field strength of 4T by Sammi et al. [16]. No T1 relaxation correction was performed for the lack of available literature values. Since brain ethanol was assumed to be distributed only within the aqueous components, concentrations were normalized to the overall water volume fraction of the VOI, as previously suggested by Hetherington et al. [12].

This procedure yielded brain ethanol concentration time courses for the four MRS measurement periods MRS1–MRS4. To assess the dynamics during the MRS periods, each ethanol concentration time course was individually modeled with a linear fit. Additionally, assuming a linear ethanol degradation during the three breaks between the MRS periods, the ends of the fitted lines were connected via straight lines. Thus, the ethanol concentration time courses could be continuously estimated throughout the two measurement sessions (~170 min).

#### 4.2.4. Comparison of Plasma Ethanol and Brain Ethanol Concentrations

Using the estimated ethanol concentration time courses from the MRS data, brain correlates were determined for the time points of the four plasma ethanol concentrations collected in between the MRS measurement periods and shortly after the last MRS measurement. A correlate of the first blood sample obtained at the end of the drinking period was not determined since the delay between the first blood sample and the start of the first MRS period was 21.9 min on average (see Table 1) and the ethanol concentration time course was considered to be highly non-linear during the resorption phase. Therefore, the linear extrapolation of the ethanol time course measured during the first MRS period would not make sense.

For a comparison with the brain ethanol concentrations (normalized to the water fraction of the brain tissue), the measured blood alcohol concentrations were normalized to the water fraction of blood.

### 4.3. Statistical Evaluation

For each subject, the relationship between the four measured plasma ethanol concentrations and their MRS brain correlates was modeled with a linear fit, which was characterized by a subject-specific slope, intercept, and correlation coefficient. Such a linear regression analysis was also performed by pooling the data from the whole subject cohort.

## Figures and Tables

**Figure 1 ijms-24-13499-f001:**
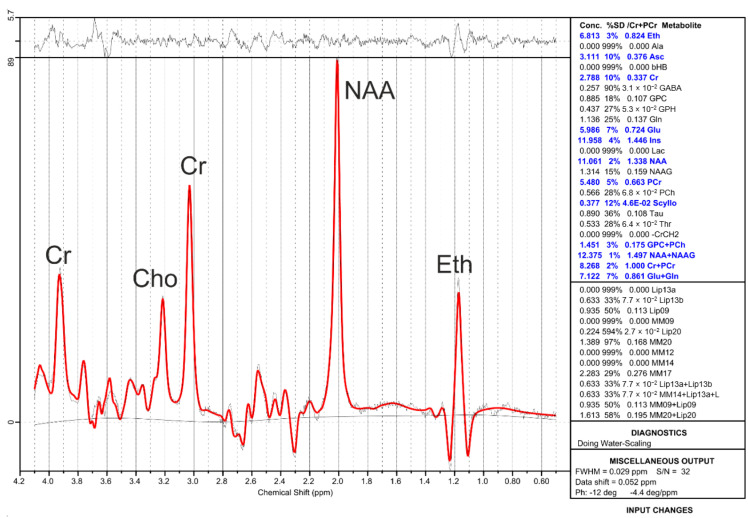
MR spectrum acquired at the beginning of the extra MRS measurement showing the ethanol resonance (Eth) as well as prominent resonances of the brain metabolites’ total N-acetyl aspartate (NAA), creatine (Cr), and choline-containing compounds (Cho). The measured spectrum is shown along with the LCModel fit (red) and quantification results (without tissue and relaxation correction), including error estimates (%SD) in the form of Cramér-Rao lower bounds (CRLBs).

**Figure 2 ijms-24-13499-f002:**
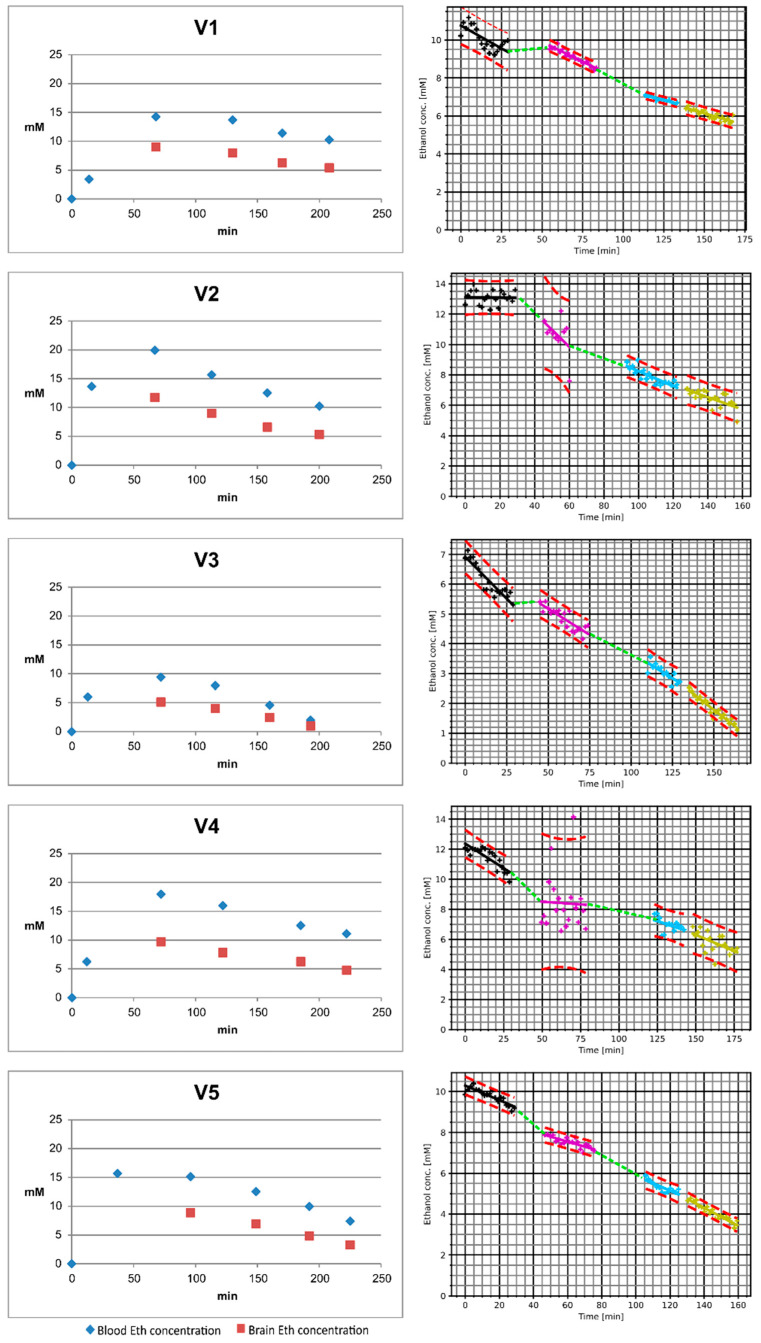
Plasma ethanol concentration values (**left**) and brain ethanol concentration curves (**right**) of the 10 study participants. Note that the time axis of the plasma ethanol concentration diagram starts with the null sample acquired right before the drinking period, while the time axis of the brain ethanol concentration diagram starts with the beginning of the first MRS session. The piecewise linear fits of the MRS measurements were linearly interpolated for the session breaks (dashed green lines). The plasma ethanol concentration diagram also contains the brain ethanol concentrations (red squares) as estimated from the brain ethanol concentration diagrams for the time points of the corresponding blood samples.

**Figure 3 ijms-24-13499-f003:**
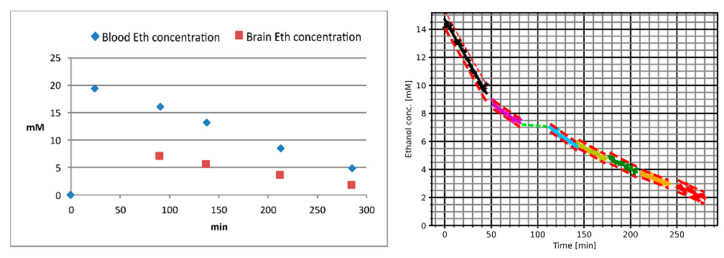
Plasma ethanol concentration values (**left**) and brain ethanol concentration curve (**right**) of the additional measurement. Note that the time axis of the plasma ethanol concentration diagram starts with the null sample acquired right before the drinking period, while the time axis of the brain ethanol concentration diagram starts with the beginning of the first MRS session. The piecewise linear fits of the MRS measurements were linearly interpolated for the session breaks (dashed green lines). The plasma ethanol concentration diagram also contains the brain ethanol concentrations (red squares) as estimated from the brain ethanol concentration diagrams for the time points of the corresponding blood samples.

**Figure 4 ijms-24-13499-f004:**
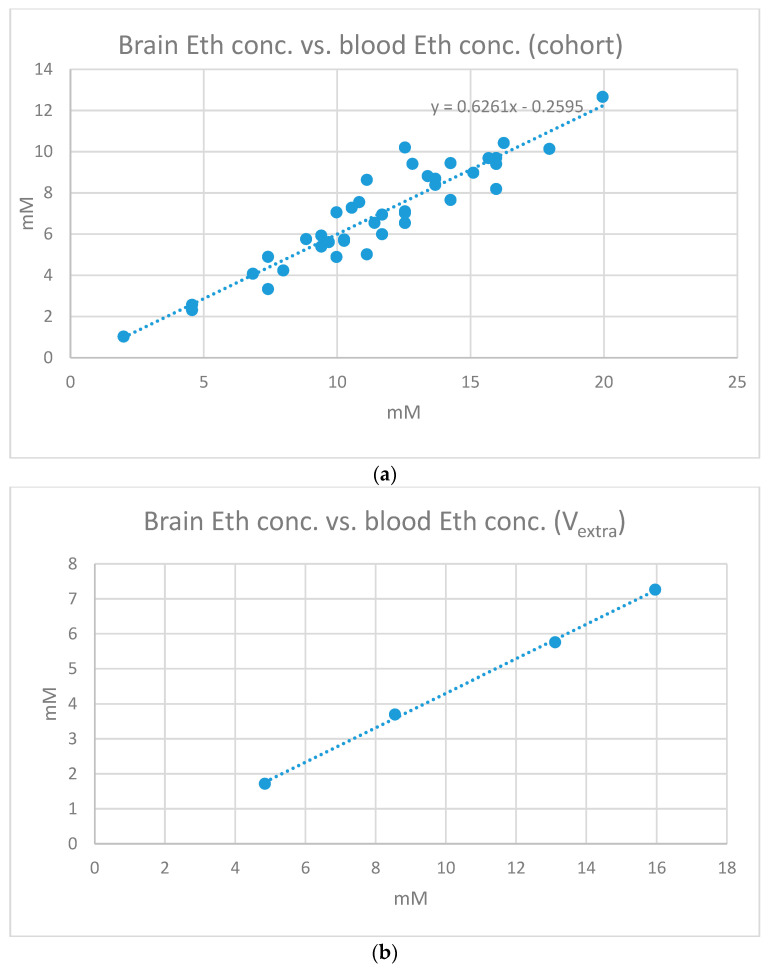
(**a**–**c**) Relationship between the brain and plasma ethanol concentrations across all ten subjects (**a**), for the additional measurement (**b**), and for V10 (**c**).

**Figure 5 ijms-24-13499-f005:**
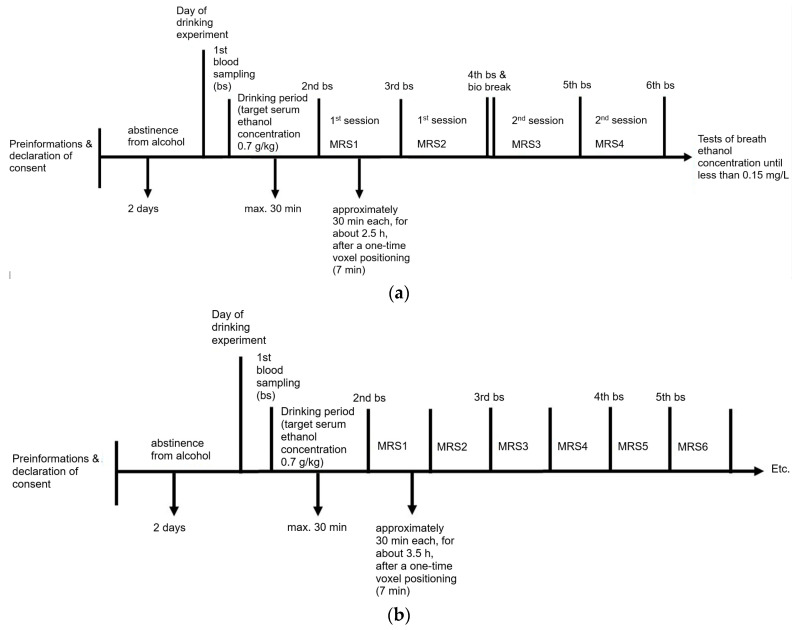
Flowchart of the cohort study (**a**) and the additional longer measurement (**b**).

**Figure 6 ijms-24-13499-f006:**
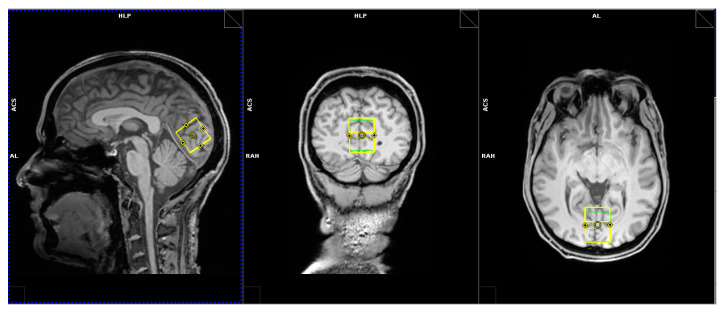
Localization (yellow box) of the MRS measurement volume (2.7 cm)^3^ in the occipital cortex.

**Table 1 ijms-24-13499-t001:** Experimental parameters.

V-Nr.	Amount of Wodka (40 vol-%) [mL]	Drinking Time [min]	Plasma Ethanol Concentration at the End of Drinking [g/L]	Interval between End of Drinking and First MRS Measurement [min]	Interval between Beginning of Drinking and First MRS Measurement [min]	Highest Measured Plasma Ethanol Concentration [g/L]	Highest Measured Brain Ethanol Concentration [mM]	Ascending Part of the Curve Observable
1	155	14	0.16	20	34	0.66	11.2	Shortly
2	200	16	0.63	21	37	0.92	13.6	No
3	95	13	0.28	26	39	0.43	7.1	No
4	153	12	0.29	33	45	0.83	14.1	No
5	116	37	0.72	25	62	0.72	10.4	No
6	137	29	0.14	27	56	0.59	9.9	Yes
7	135	26	0.95	19	45	0.95	12.0	(Not available)
8	95	15	0.13	19	34	0.5	7.9	Yes
9	100	17	0.45	14	31	0.75	11.3	No
10	160	6	0.04	15	21	0.74	12.2	Yes

**Table 2 ijms-24-13499-t002:** Linear regression and correlation analysis for the subject cohort: Pearson’s correlation coefficients as well as the slope and intercept of the linear regression are listed for the individual subjects, including the cohort means.

	Pearson’s Corr. Coeff.	Slope	Intercept
V01	0.9909	0.9028	−3.6794
V02	0.9989	0.7264	−1.8130
V03	0.9969	0.5688	−0.1000
V04	0.9884	0.6953	−2.5368
V05	0.9982	0.7434	−2.3142
V06	0.9737	0.9330	−2.2611
V07	0.9862	0.5600	0.7960
V08	0.9997	0.8402	−1.5809
V09	0.9987	0.7452	−1.6414
V10	0.4773	0.4880	1.7454
	0.9409 ± 0.1631	0.7203 ± 0.1481	−1.3385 ± 1.6565

**Table 3 ijms-24-13499-t003:** Personal data and heath information of the 10 participants (m: male, f: female).

V-Nr.	Sex	Age [a]	Body Weight [kg]	Body Height [cm]	BMI	Ethanol Uptake per Week (Average)	Known Illnesses	Regular Medication
1	M	35	80	178	25.2	0.5 l red wine	None	None
2	M	61	112	194	29.8	0.25 l beer	Hemithyreoidektomie	None
3	F	29	58	168	20.5	0.1 l wine	None	None
4	M	33	80	186	23.1	1 l beer	Coeliac disease	None (gluten-free diet)
5	F	28	65	165	23.9	0.5 l wine	None	None
6	M	67	72	176	23.2	1 l wine	Arterial hypertension	Ramipril
7	F	60	90	154	37.9	Very low	Arterial hypertension	Ramipril, hydrochlorothiazide, amlodipine
8	F	29	57	160	22.3	0.25 l wine	None	None
9	F	63	60	168	21.3	0.25 l wine	Arterial hypertension	Spironolactone, cholecalciferol
10	M	61	84	187	24	Very low	Tinnitus	Testosterone

## Data Availability

Data are fully demonstrated in the article. Row data can be requested from the authors.

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
