# Peer review of "Ethanol Kinetics in the Human Brain Determined by Magnetic Resonance Spectroscopy"

_ijms, 2023, doi:10.3390/ijms241713499_

Round 1
Reviewer 1 Report (Previous Reviewer 1)
To the Authors,
In this paper, the Authors are studying ethanol in the blood and in the brain, in particular the occipital cortex using a single voxel spectroscopy approach at 3T. Their focus is (i) to check that “brain ethanol kinetics closely follow the serum ethanol curve” and (ii) to investigate the NMR visibility of ethanol as it has been debated in the community. A rather disparate cohort of ten volunteers have been examined for 4 or more MRS sessions following the absorption of a substantial quantity of ethanol, with blood being drawn in-between those NMR session for dosage of ethanol in the blood. The results validate a rapid biodistribution of ethanol and stable equilibrium between the vascular and brain tissue compartments. In addition, the observation of a ratio of approximately 63% between the estimated brain and blood ethanol levels goes along the hypothesis that a large fraction of brain ethanol could be NMR-invisible.
The Authors have improved their manuscript and answered some of my previous remarks. Yet I remain unconvinced by few of their methodological arguments mainly regarding their estimation of ethanol concentration and its NMR invisibility. Additional remarks have been added below the Authors’s responses.
List of major issues/comments:
Q1. As stated before, ethanol is subjected to a substantial magnetization/saturation transfer effect as demonstrated by Fein and Meyerhoff [PMID: 10968662]. This is the likely reason for the apparent brain ethanol concentration being 63% of the blood ethanol levels.
R1. We are aware of the magnetization transfer that may affect the ethanol visibility for cer-tain sequences. However, this effect was observed when using dedicated off- or on-resonance saturation schemes (Fein and Meyerhoff). In this paper, a CHESS water suppression method was used and the authors claim that it has a negligible MT effect. It should be noted that the WET water suppression scheme used in our study is very similar to this CHESS scheme. It also consists of three spectrally selective pulses with spoiler gradients. In our measurements the water suppression pulses had a (quite small) bandwidth of 50 Hz. We therefore expect that MT effects arising from water suppression are also negligible for our study. We included the details of water suppression in the revised manuscript to clarify this issue. The review implies that with an optimized sequence the invisible ethanol pool can be made visible so that overall ethanol visibility can be increased to nearly 100%. In this point, we disagree: By avoiding MT effects, you can only make sure that the visibility of the mobile ethanol pool is not further reduced, but the invisible pool remains invisible. If there is a substantial ethanol pool bound in cell membranes (as suggested in the literature), it is unaccessible by the methods presented in all these papers or suggested by the reviewer (metabolite cycling). The (comparably high) ethanol visibility of approximately 100 % reported by Hetherington et al. may be due to other factors (e.g., different sequence parameters, quantification methods and relaxation correction) and just shows that the ethanol visibility is still an unresolved and controversial issue.
While I am not disputing the fact that the NMR signal from most neurometabolites would not be significantly impacted by the WET water suppression scheme, there is no evidence that it is the same for ethanol. Ethanol could very well be in relatively fast chemical exchange or dipolar coupling with water leading to significant saturation/magnetization transfer effects.
This argument about the sensitivity of ethanol to MT is directly coupled to the unresolved issue of its NMR (in)visibility. Indeed one could hypothesize the existence of pools of ethanol interacting with water or membrane. Ethanol in those pools would be difficult to detect using NMR due to its assumed sensitivity to ST/MT/NOE effects and its drastically shorten effective T2 and T1 relaxation times. Therefore, one would ideally use a sequence that minimizes the amount of radiofrequency and work at very short echo-time and long TR.
That is not the case for the approach adopted by the Authors. Therefore, I would advise them to discuss this point along the other limitations of this study.
Q2. No details are given in the Materials and Methods section regarding the water presaturation module used in conjunction with the semi-LASER localization sequence. What was the typical chemical shift displacement artefact (CSDA) between the ethanol resonance and the water signal used as a reference? Was a frequency offset used to minimize this CSDA?
R2. Details of the water suppression module have been included in the revised manuscript. For the actual (water-suppressed) measurement, we set the center frequency of the localization pulses to the frequency of the ethanol methyl resonance (1.2 ppm) while in the water reference scan that center frequency was set to 4.7 ppm. With the 5 kHz sLASER localization pulses the CSDA is quite small anyway. A sentence was added to the methods section.
Q3. Line 352: As LCModel can only give positive values, one would expect the intercept to be positive since positive EtOH level could be detected even in absence of EtOH in the blood (but not the opposite). Also there is an argument that brain EtOH should be negligible (null) in absence of EtOH in the blood. Therefore, it would be reasonable to force the linear regression to have a intercept at (0,0).
R3. LCModel only gives positive concentration values, but a quantification bias can go in either direction, depending on the baseline fit and superimposed resonances from other metabolites (not only at 1.2. ppm, but also at 3.7 ppm). MRS data before ethanol update were not acquired in this study. Note that Fig. 6b shows almost perfect linear regression (r = 0.9994) with a substantial y-axis offset. This suggests that the quantification bias of ethanol might be concentration-dependent.
List of minor issues/comments:
Q4. How was the LCModel basis sets simulated?
R4. The LCModel basis-set was simulated with the PyGAMMA library from the VeSPA pack-ge (Soher et al., Magn Reson Med. 2023 Sep;90(3):823-838).
Q5. The TR is rather short at 1.5 second. How did the Authors accounted for the differential T1-weightings between metabolites and the water reference?
R5. We did not find a T1 value for ethanol at 3T in the literature. Since T1 values between 1000 and 1500 ms are reported for all major MR-visible brain metabolites at 3T (including water in GM and WM), we assume that the ethanol T1 probably lies within this range, too. A TR = 1.5 then only gives very moderate correction factors between 0.95 and 1.05, which can be neglected compared to the non-negligible T2 correction.
I am not convinced. First, the spectroscopic voxel is located in the occipital cortex across both hemispheres encompassing some cerebrospinal fluid that possess a T1 of about 4 to 5 s. Besides as stated previously it is likely that ethanol exists in different physicochemical states in the brain with very short to relatively long effective relaxation times. In this context, acquiring spectroscopic data using a sequence with a long TE and a short TR constitute a major obstacle to any attempt at an absolute quantification. This limitation should be discussed and any conclusion regarding the NMR visibility of ethanol should be prudent.
Quality of English is fine.
Author Response
List of major issues/comments:
Q1. As stated before, ethanol is subjected to a substantial magnetization/saturation transfer effect as demonstrated by Fein and Meyerhoff [PMID: 10968662]. This is the likely reason for the apparent brain ethanol concentration being 63% of the blood ethanol levels.
R1. We are aware of the magnetization transfer that may affect the ethanol visibility for cer-tain sequences. However, this effect was observed when using dedicated off- or on-resonance saturation schemes (Fein and Meyerhoff). In this paper, a CHESS water suppression method was used and the authors claim that it has a negligible MT effect. It should be noted that the WET water suppression scheme used in our study is very similar to this CHESS scheme. It also consists of three spectrally selective pulses with spoiler gradients. In our measurements the water suppression pulses had a (quite small) bandwidth of 50 Hz. We therefore expect that MT effects arising from water suppression are also negligible for our study. We included the details of water suppression in the revised manuscript to clarify this issue. The review implies that with an optimized sequence the invisible ethanol pool can be made visible so that overall ethanol visibility can be increased to nearly 100%. In this point, we disagree: By avoiding MT effects, you can only make sure that the visibility of the mobile ethanol pool is not further reduced, but the invisible pool remains invisible. If there is a substantial ethanol pool bound in cell membranes (as suggested in the literature), it is unaccessible by the methods presented in all these papers or suggested by the reviewer (metabolite cycling). The (comparably high) ethanol visibility of approximately 100 % reported by Hetherington et al. may be due to other factors (e.g., different sequence parameters, quantification methods and relaxation correction) and just shows that the ethanol visibility is still an unresolved and controversial issue.
While I am not disputing the fact that the NMR signal from most neurometabolites would not be significantly impacted by the WET water suppression scheme, there is no evidence that it is the same for ethanol. Ethanol could very well be in relatively fast chemical exchange or dipolar coupling with water leading to significant saturation/magnetization transfer effects.
This argument about the sensitivity of ethanol to MT is directly coupled to the unresolved issue of its NMR (in)visibility. Indeed one could hypothesize the existence of pools of ethanol interacting with water or membrane. Ethanol in those pools would be difficult to detect using NMR due to its assumed sensitivity to ST/MT/NOE effects and its drastically shorten effective T2 and T1 relaxation times. Therefore, one would ideally use a sequence that minimizes the amount of radiofrequency and work at very short echo-time and long TR.
That is not the case for the approach adopted by the Authors. Therefore, I would advise them to discuss this point along the other limitations of this study.
>> We included a discussion about this issue in the limitation paragraph.
Q2. No details are given in the Materials and Methods section regarding the water presaturation module used in conjunction with the semi-LASER localization sequence. What was the typical chemical shift displacement artefact (CSDA) between the ethanol resonance and the water signal used as a reference? Was a frequency offset used to minimize this CSDA?
R2. Details of the water suppression module have been included in the revised manuscript. For the actual (water-suppressed) measurement, we set the center frequency of the localization pulses to the frequency of the ethanol methyl resonance (1.2 ppm) while in the water reference scan that center frequency was set to 4.7 ppm. With the 5 kHz sLASER localization pulses the CSDA is quite small anyway. A sentence was added to the methods section.
Q3. Line 352: As LCModel can only give positive values, one would expect the intercept to be positive since positive EtOH level could be detected even in absence of EtOH in the blood (but not the opposite). Also there is an argument that brain EtOH should be negligible (null) in absence of EtOH in the blood. Therefore, it would be reasonable to force the linear regression to have a intercept at (0,0).
R3. LCModel only gives positive concentration values, but a quantification bias can go in either direction, depending on the baseline fit and superimposed resonances from other metabolites (not only at 1.2. ppm, but also at 3.7 ppm). MRS data before ethanol update were not acquired in this study. Note that Fig. 6b shows almost perfect linear regression (r = 0.9994) with a substantial y-axis offset. This suggests that the quantification bias of ethanol might be concentration-dependent.
List of minor issues/comments:
Q4. How was the LCModel basis sets simulated?
R4. The LCModel basis-set was simulated with the PyGAMMA library from the VeSPA pack-ge (Soher et al., Magn Reson Med. 2023 Sep;90(3):823-838).
Q5. The TR is rather short at 1.5 second. How did the Authors accounted for the differential T1-weightings between metabolites and the water reference?
R5. We did not find a T1 value for ethanol at 3T in the literature. Since T1 values between 1000 and 1500 ms are reported for all major MR-visible brain metabolites at 3T (including water in GM and WM), we assume that the ethanol T1 probably lies within this range, too. A TR = 1.5 then only gives very moderate correction factors between 0.95 and 1.05, which can be neglected compared to the non-negligible T2 correction.
I am not convinced. First, the spectroscopic voxel is located in the occipital cortex across both hemispheres encompassing some cerebrospinal fluid that possess a T1 of about 4 to 5 s. Besides as stated previously it is likely that ethanol exists in different physicochemical states in the brain with very short to relatively long effective relaxation times. In this context, acquiring spectroscopic data using a sequence with a long TE and a short TR constitute a major obstacle to any attempt at an absolute quantification. This limitation should be discussed and any conclusion regarding the NMR visibility of ethanol should be prudent.
>> We would like to point out that a TR=1500 ms is not particularly short, but a commonly used setting in brain MRS studies performed at 3T. Note that Sammi et al. and Hetherington et al. used a TR=2 s at 4T where the T1 of water and metabolites is even longer than at 3T. Furthermore, it should be noted that a T1 bias should be smaller for ethanol than for other major brain metabolites, which are only present in GM and WM, but not in CSF. Ethanol is also present in CSF and due to its excellent water solubility the ethanol concentration scales with the tissue water fraction. Therefore, in terms of the T1 bias, the tissue composition (GM/WM/CSF) will affect the quantification results of the ethanol signal and the water reference signal in a similar way since it can be assumed that the ethanol T1 in CSF is much longer that the ethanol T1 in GM and WM (as is the case for the water T1). We included a paragraph about this in the limitations section of the discussion.
Reviewer 2 Report (Previous Reviewer 2)
authors have adequately responded to previous comments.
Author Response
Thank you very much for your dealing with our manuscript and the resulting improvement!
Reviewer 3 Report (New Reviewer)
In this study, the correlation between brain and blood ethanol concentrations was examined.
The number of participants, as highlighted by the authors, unfortunately is limited, and some inter-individual variations in ethanol kinetics emerged without a clear explanation. Nevertheless, a strong correlation between Blood Alcohol Concentration (BAC) and blood ethanol levels was observed, which, in my opinion, represents the most significant finding. The implications of these results in the forensic context, which is relevant to many of the authors, should be discussed more extensively.
This finding confirms the validity of using BAC to reflect the direct effects of ethanol on the brain. Do the authors suggest the potential use of spectroscopic techniques for diagnosing acute alcohol intoxication? What would be the advantages of such an approach? What are the current limitations of this diagnostic technique? Could it be related to execution times or the impracticality of conducting on-site tests due to the need for expensive and non-portable diagnostic instruments?
Is it conceivable to apply this technique in the future? If so, how many years might it take?
Please remove the red underlines (automatic spelling checker) in Figure 1.
Author Response
>> Thank you very much for your dealing with our article and the interesting points of discussion. This helps to focus on the main hypothesis and emphasize the relevant findings. Thank you very much for the comments that very much contribute to an improvement of this manuscript.
In this study, the correlation between brain and blood ethanol concentrations was examined.
The number of participants, as highlighted by the authors, unfortunately is limited, and some inter-individual variations in ethanol kinetics emerged without a clear explanation. Nevertheless, a strong correlation between Blood Alcohol Concentration (BAC) and blood ethanol levels was observed, which, in my opinion, represents the most significant finding. The implications of these results in the forensic context, which is relevant to many of the authors, should be discussed more extensively.
>>This is emphasized in the discussion.
This finding confirms the validity of using BAC to reflect the direct effects of ethanol on the brain. Do the authors suggest the potential use of spectroscopic techniques for diagnosing acute alcohol intoxication? What would be the advantages of such an approach? What are the current limitations of this diagnostic technique? Could it be related to execution times or the impracticality of conducting on-site tests due to the need for expensive and non-portable diagnostic instruments?
Is it conceivable to apply this technique in the future? If so, how many years might it take?
>>The implications of these study results on the future diagnostic approaches are added to the discussion.
Please remove the red underlines (automatic spelling checker) in Figure 1.
>> In the current formatting not applicable.
Reviewer 4 Report (New Reviewer)
The authors present an experimental study on 10 volunteers (5 M and 5F) comparing brain (measured through in vivo MRS) and blood (measured through HS-GC-FID) ethanol concentrations after an ingestion of ethanol (Eth) calculated using the Widmark’s equation to reach a standardized blood alcohol concentration (BAC) of 0,92 g/L, corresponding to 0.7 g/kg.
The paper is well written, the authors have used a rigorous methodological approach, with several measurements in a long period of observation. The reported data showed a strong intra individual correlation between plasma ethanol concentration value and brain ethanol concentration curves, suggesting a rapid equilibration between blood and brain.
I retain that such type of studies (Hetherington et al., Kubo et al, Fein et al) may be really useful to the scientific community both from a clinical and a forensic point of view (i.e. some “experts” often claim at trial that only brain Eth concentration exactly depicts the impairment of the individual).
The main limitation of the study is the selected cohort of investigation. Ten (10) volunteers are a small sample to drive any definite conclusion, but the authors recognize this at the end of the manuscript. Moreover, the authors have underlined that the heterogeneity of the subject cohort (sexes and large age range) could be a limit (the obtained MRS brain ethanol curves showed distinct inter-individual differences). Anyway, despite this limit, as predicted, the results showed a strong correlation between plasma and brain eth concentrations.
Finally, a critical point refers to the MRS visibility of ethanol (only 63% of blood ethanol). This might be due to a water suppression problem or more probably to the fact that there is a substantial ethanol pool bound in cell membranes; it would be important to discuss this latter hypothesis citing previous literature and adding some molecular speculations.
no comment
Author Response
The authors present an experimental study on 10 volunteers (5 M and 5F) comparing brain (measured through in vivo MRS) and blood (measured through HS-GC-FID) ethanol concentrations after an ingestion of ethanol (Eth) calculated using the Widmark’s equation to reach a standardized blood alcohol concentration (BAC) of 0,92 g/L, corresponding to 0.7 g/kg.
The paper is well written, the authors have used a rigorous methodological approach, with several measurements in a long period of observation. The reported data showed a strong intra individual correlation between plasma ethanol concentration value and brain ethanol concentration curves, suggesting a rapid equilibration between blood and brain.
I retain that such type of studies (Hetherington et al., Kubo et al, Fein et al) may be really useful to the scientific community both from a clinical and a forensic point of view (i.e. some “experts” often claim at trial that only brain Eth concentration exactly depicts the impairment of the individual).
The main limitation of the study is the selected cohort of investigation. Ten (10) volunteers are a small sample to drive any definite conclusion, but the authors recognize this at the end of the manuscript. Moreover, the authors have underlined that the heterogeneity of the subject cohort (sexes and large age range) could be a limit (the obtained MRS brain ethanol curves showed distinct inter-individual differences). Anyway, despite this limit, as predicted, the results showed a strong correlation between plasma and brain eth concentrations.
Finally, a critical point refers to the MRS visibility of ethanol (only 63% of blood ethanol). This might be due to a water suppression problem or more probably to the fact that there is a substantial ethanol pool bound in cell membranes; it would be important to discuss this latter hypothesis citing previous literature and adding some molecular speculations.
>> We included a discussion about this issue in the limitation paragraph of the revised manuscript.
This manuscript is a resubmission of an earlier submission. The following is a list of the peer review reports and author responses from that submission.
Round 1
Reviewer 1 Report
In this paper, the Authors are comparing the kinetics of ethanol in the blood and in the brain, in particular the occipital cortex using a single voxel spectroscopy approach at 3T. Their focus is (i) to check that “brain ethanol kinetics closely follow the serum ethanol curve” and (ii) to investigate the NMR visibility of ethanol as it has been debated in the community. A rather disparate cohort of ten volunteers have been examined for 4 or more MRS sessions following the absorption of a substantial quantity of ethanol, with blood being drawn in-between those NMR session for dosage of ethanol in the blood. The results validate a rapid biodistribution of ethanol and stable equilibrium between the vascular and brain tissue compartments. In addition, the observation of a ratio of approximately 63% between the estimated brain and blood ethanol levels goes along the hypothesis that a large fraction of brain ethanol could be NMR-invisible.
While the methodology is suitable for most in vivo 1H NMR studies, the Authors have ignored the fact that ethanol is subjected to a large magnetization/saturation transfer effect (of about 30%) as demonstrated by Fein and Meyerhoff [PMID: 10968662]. Consequently, they should have conducted their study without the “assumed” use of a typical water suppression scheme (no details given in the Materials and Methods section). This was the motivation for the use of an numerically optimized frequency selective refocusing pulse by Hetherington et al [10] that concluded on a 100% NMR visibility of ethanol in the brain. An interesting alternative would have been to adopt a “metabolic cycling” approach (DOI: 10.1002/mrm.26873). This is a major flaw that cannot be solved a posteriori which is quite unfortunate.
List of major issues/comments:
Q1. As stated before, ethanol is subjected to a substantial magnetization/saturation transfer effect as demonstrated by Fein and Meyerhoff [PMID: 10968662]. This is the likely reason for the apparent brain ethanol concentration being 63% of the blood ethanol levels.
Q2. No details are given in the Materials and Methods section regarding the water presaturation module used in conjunction with the semi-LASER localization sequence. What was the typical chemical shift displacement artefact (CSDA) between the ethanol resonance and the water signal used as a reference? Was a frequency offset used to minimize this CSDA?
Q3. Line 352: As LCModel can only give positive values, one would expect the intercept to be positive since positive EtOH level could be detected even in absence of EtOH in the blood (but not the opposite). Also there is an argument that brain EtOH should be negligible (null) in absence of EtOH in the blood. Therefore, it would be reasonable to force the linear regression to have a intercept at (0,0).
List of minor issues/comments:
Q4. How was the LCModel basis sets simulated?
Q5. The TR is rather short at 1.5 second. How did the Authors accounted for the differential T1-weightings between metabolites and the water reference?
The writing is often confusing with many convoluted sentences. The help of a native English writer/editor should be sought after.
Reviewer 2 Report
This paper is interesting in having both in vivo MRS measures of ethanol (EtOH and repeated plasma EtOH concentration measures collected at the same time as the scan in humans. The conclusion, that EtOH kinetics are unique to individuals, is not novel. It is well known that EtOH kinetics are dependent on a variety of factors, including age, sex, race, and body composition.
One major concern with the manuscript is the inclusion of only 10 individuals with a wide age range (28 – 67 years), a wide BMI range (20.5 – 37.9), and a wide range of weekly drinking (0.1 – 1L). In addition, 3 of the 10 participants had arterial hypertension for which they were medicated.
While one could argue that the study is meant to demonstrate that equivalent EtOH doses result in differing kinetics because of age, sex, BMI, etc, another major issue with this study is that the experimental conditions were not well controlled. The volume (ml) of EtOH (vodka) given to each of the 10 participants ranged from 95 – 200ml, the time taken to consume EtOH ranged from 6 – 37 min, and the interval between end of drinking and MRS scan ranged from 21 – 62min.
Another major concern is that the MRS estimate of EtOH was only 63% of the plasma EtOH concentration. As other studies have reported upwards of 90% of EtOH in the brain corresponding with blood EtOH estimates, the quality of the spectra acquired in this study are called into question.
No justification was given for using an occipital cortex voxel.
Participants were given a light meal 2h before exposure to EtOH, but time of last meal and calories consumed at that meal were not considered.
The authors do not cite the rich literature using alcohol clamps to describe EtOH kinetics in humans, which was demonstrated to be better than oral EtOH exposure for alcohol challenge studies (e.g., Ramachandani, 2001). Further, other relevant work is not referenced (e.g., Gomez, 2011; Monnig 2019; Tunc-Skarka 2015; Kubo 2013).
In figure 6, please clarify the numerous points when only 10 individuals had data collected? If multiple measures are represented, how many per participant?
Please check for formatting (e.g., sometimes "M" sometimes "m" were used in Table 1). Also other issues with grammar, word choice, and typos.